# Thermal optimized PCR coupled to CRISPR-Cas12a for rapid detection of bla$_{OXA-1}$ resistance gene

Ana Quiroz-Huanca[1,2], Maryhory Vargas-Reyes[1], Juan Diego López[1,2], Kiara Flores-Jimenez[1,2], Sofia Saldarriaga-Morán[1,2], Karla Cifuentes[1,2], Roberto Alcántara[1,3]*

1 Laboratory of Biomolecules, Faculty of Health Sciences, Universidad Peruana de Ciencias Aplicadas (UPC), Lima, Peru, 2 School of Biology, Faculty of Health Sciences, Universidad Peruana de Ciencias Aplicadas (UPC), Lima, Peru, 3 School of Biosciences and Veterinary Medicine, University of Camerino, Camerino, Italy

* roberto.alcantara@upc.pe

## Abstract

The β-lactams are critically important broad-spectrum antibiotics, widely used as first-line treatments; however, their effectiveness is increasingly compromised by β-lactamase enzymes. Among these, OXA-type enzymes have expanded to over 400 variants and are highly prevalent in *Enterobacteriaceae*. Current phenotypic and molecular detection tests have long turnaround times or require specialized equipment, respectively. In this study, we optimize a rapid molecular assay combining a PCR with modified thermal ramp rate (TRR) along with CRISPR-Cas12a fluorescence detection for bla$_{OXA-1}$-harboring *E. coli* isolates. Using a commercial DNA Taq polymerase (TRR: 2.2 °C/s, annealing and extension hold time: 1 s), amplification time was reduced from 80 to 30 min, enabling detection within 50 min (PCR: 30 min; CRISPR: 20 min). With a locally produced enzyme (hold: 10 s), amplification time was 44 min. To demonstrate the practical application of the assay, we evaluated spiked poultry fecal samples achieving an analytical sensitivity of 8 CFU/reaction using commercial DNA Taq polymerase. The accelerated PCR:CRISPR workflow delivers results in less than one hour without compromising technical sensitivity (attomoles range), not requiring high technical expertise, and can be implemented in laboratories with basic molecular biology equipment.

## Introduction

Antimicrobial resistance (AMR) is recognized as one of the most pressing global public health challenges, as it undermines the efficacy of antibiotic-based treatments and contributes to increased mortality from both community- and hospital-acquired infections. In 2021, it was estimated that approximately 4.71 million deaths were associated with AMR, with 1.14 million deaths directly attributable to it [1]. Among the

**Data availability statement:** All relevant data are within the manuscript and its Supporting Information files.

**Funding:** "This research was supported by an internal research grant from the Universidad Peruana de Ciencias Aplicadas with number C-004-2021-1 awarded to RA. There was no additional external funding received for this study. The funder had no role in study design, data collection and analysis, decision to publish, or preparation of the manuscript.".

**Competing interests:** The authors declare that they do not have competing interests.

diverse mechanisms of antibiotic resistance, β-lactamases are of particular concern due to their ability to hydrolyze penicillin, carbapenem, and even some β-lactam/β-lactamase inhibitor combinations [2]. OXA-type β-lactamases, including $bla_{OXA-1}$ and its variants, encode oxacillinases that confer resistance primarily to ampicillin, and cephalosporins [3]. Notably, blaOXA-1 has been reported as one of the most prevalent blaOXA groups in clinical *Enterobacteriaceae* isolates globally [4,5].

Currently, diverse methods could be used to detect $bla_{OXA}$ genes and their associated antibiotic susceptibility. Among phenotypic antimicrobial susceptibility tests (AST), the standard disk diffusion and microdilution assays are commonly used, including modified protocols and commercial options [6–8]. While phenotypic methods are cost-effective and simple to use, their diagnostic accuracy varies depending on the β-lactamase type and the bacteria host species, especially for Gram-negative bacteria [8,9].

Molecular methods, beyond DNA sequencing, for detecting $bla_{OXA}$ variants include nucleic acid amplification techniques such as conventional PCR [10–13], and isothermal amplification via loop-mediated assay (LAMP) [14,15]. Despite their technical advantages, molecular tests and sequencing remain limited in low-resource settings due to high implementation costs and restricted access to related consumables [16,17]. Currently, CRISPR-Cas systems have emerged as promising molecular tools for nucleic acid detection. Among them, Cas12a is notable for its collateral cleavage activity, which enables detection using labeled probes that allow a myriad of readout approaches [18–20]. Several studies have applied CRISPR-Cas technology for detecting bacterial pathogens [21], and others for detecting antibiotic resistance genes (ARG), including β-lactamases [22,23].

Even with the competitive detection performance of CRISPR-Cas assays, most of them rely on a pre-amplification step [24]. Conventional PCR, a common pre-amplification method, relies on repeating three sequential steps of different temperatures to amplify a specific target needing long amplification times. Previous reports have introduced variations to conventional PCR by modifying ramp rates and reducing holding times to shorten amplification cycles [25,26]. In line with this evidence, the present study reports a proof-of-concept for a PCR with a modified thermal cycling protocol combined with CRISPR-Cas for the rapid fluorescent detection of $bla_{OXA-1}$-harboring *E.coli* isolates. By utilizing spiked poultry feces as a model, we demonstrate the assay's potential for efficient monitoring of antimicrobial resistance genes in complex matrices using basic molecular biology equipment. This dual approach substantially reduced the time-to-results while maintaining competitive analytical sensitivity.

## Materials and methods

### Oligos design

A total of 46 sequences corresponding to the $bla_{OXA-1}$ family were obtained from the NDARO and the CARD database (S1 File in S1 Text). The sequences were aligned using MUSCLE on the Unipro UGENE platform (version 51.0) [27] to identify a conserved region for designing the crRNA and primers, which was done using the CHOPCHOP [28] and CRISPRScan [29] programs. For crRNA prediction in both

platforms, parameters were used as default for the LbCas12a with a TTTN PAM. The candidate sequences obtained from each program were consolidated. Those candidates that aligned with the sequence of the primers were discarded. For the top 10 candidate sequences, secondary structures were predicted using the ViennaRNA Web Services, with temperature et to 25°C and the NaCl concentration set to 50 mM in the energy parameter options [30]. Those sequences that showed structures with a ΔG < −5.00 kcal/mol, and a %GC >= 25% were selected. Sequence specificity was assessed by alignment against other $bla_{OXA}$ groups (S2 File in S1 Text), using the Unipro UGENE platform (version 51.0). Finally, the candidate with a sequence 5'-ACCAGAAGAACAAAUUCAAUUCCU-3,' which recognizes a sequence adjacent to a PAM – TTTC, was selected. Then the LbCas12a-scafold sequence (5'-UAAUUUCUACUAAGUGUAGAU-3') and the T7 promoter (5'-TAATACGACTCACTATAGG-3') were added 5'-upstream. Likewise, the primers used to amplify the target region were designed in the CHOPCHOP program, forward 5'- AGCATGGCTCGAAAGTAGCTTA-3,' and reverse 5'-GACCCCAAGTTTCCTGTAAGTG-3' [28].

## Target amplification and Optimization of PCR thermal cycling

A 282 bp region of the $bla_{OXA-1}$ gene from the genomic DNA of an *E. coli* isolate, previously characterized by whole genome sequencing [31], was amplified with conventional or modified PCR protocol. Reactions were prepared using the DreamTaq Green PCR Master Mix (2X) (Cat. N° K1081, ThermoFisher), in a final volume of 20 µL, including 0.2 µM of each primer, and 10 ng of genomic DNA. A conventional PCR reaction was included as a positive control using the same reagents and DNA target.

Conventional PCR was performed as follows: initial denaturation at 95 °C for 3 min, followed by 30 cycles of 95 °C, 60 °C, and 72 °C for 30 s each, with a final extension at 72 °C for 5 min. Thermal ramp rate (TRR) was set to 0.5 °C/s, with a total amplification time of 82 minutes. For the modified PCR, the same conditions of initial denaturation and final elongation were used with cycles of 95°C and 60°C for 1 s each. Different TRR were evaluated, including 0.6 °C/s (70 min), 0.8 °C/s (56 min), 1.2 °C/s (40 min), 1.6 °C/s (34 min), and 2.2 °C/s (30 min). All reactions were carried out on the T100 Thermal Cycler (BIO-RAD, USA). The amplification products were visualized by electrophoresis (80 V for 45 min) in agarose gel (1.7%), using a pre-staining method by mixing the tracking buffer with SYBR Gold (Cat. N° S11494, ThermoFisher).

The modified PCR protocol was also evaluated with a DNA Taq polymerase produced in the laboratory [31]. PCR reactions were prepared in a final volume of 20 µL, including 0.5 mM dNTPs, 2 ng/µL DNA Taq polymerase, 0.2 µM each primer, and 10 ng genomic DNA, in 1X amplification buffer (50 mM Tris-HCl, 75 mM KCl, 3 mM MgCl2, 10% trehalose, 10 mM DTT, 0.1 mM EDTA, pH: 8.4, 25 °C). Different time intervals were evaluated for the amplification cycles, including 30 s, 15 s, 10 s, 5 s, and 1 s. Once the shortest time interval was defined, different TRR values were evaluated, including 0.6 °C/s (83 minutes), 0.8 °C/s (69 minutes), 1.2 °C/s (54 minutes), 1.6 °C/s (47 minutes), and 2.2 °C/s (44 minutes). The amplification products were visualized by electrophoresis (80 V for 45 min) in agarose gel (1.7%), using a pre-staining method by mixing the tracking buffer with SYBR Gold.

## CRISPR-Cas reaction

An LbCas12a produced in the laboratory was used for the CRISPR-Cas reaction [32], following previous standardized reaction condition [33,34]. Initially, an aliquot of the crRNA was refolded at 65°C for 10 min, followed by incubation at room temperature for the same time. The crRNA:Cas complex was prepared in 10 µL of 1X NEBuffer r2.1 (50 mM NaCl, 10 mM Tris-HCl, 100 µg/mL BSA, pH 7.9–25°C), mixing 10 nM LbCas12a, 15 nM crRNA, and 200 nM of the ssDNA 5'-5FAM-TTATT-BHQ1–3' probe. The complex was then incubated at room temperature for 10 min. In parallel, 5 µL of the amplification products were diluted in 85 µL of 1X NEBuffer r2.1 supplemented with 20 mM $MgCl_2$. The fluorescence reading was performed on a 96-well flat-bottom black microplate (Cat. No. 23710, ThermoFisher) using a Synergy H1 (Agilent, USA) or Varioskan LUX (ThermoFisher, USA) plate multireader with an excitation wavelength at 491 nm and emission at 525 nm. Comparison between multireader instruments was performed by measuring fluorescence of a positive and negative controls (S1 Fig in S1 Text). The recorded fluorescence values were normalized ($NF_{ntc}$) by calculating a ratio between the

fluorescence of the reactions of the samples divided by the fluorescence of the non-template control reaction (NTC). The assay specificity was evaluated against a set of *E,coli* isolates harboring other beta-lactamases, including $bla_{CMY}$, $bla_{CTX15}$, $bla_{CTX65}$, $bla_{TEM1b}$, and $bla_{TEM176}$.

## Analytical performance

**Limit of blank (LoB).** To determine the cutoff point of the assay, LoB was calculated, using 10 $bla_{OXA-1}$ negative *E. coli* isolates that were previously characterized by whole genome sequencing [31]. The genomic DNA was amplified, and the amplification products were detected using CRISPR-Cas. For the calculation of the LoB, the mean and standard deviation (SD) of the $NF_{ntc}$ values were calculated. Then the LoB was calculated according to the formula

$LoB = NF_{ntc}\ mean + 2 \times NF_{ntc}\ SD$, for values observed at 20 min [35].

**Target concentration curve.** To determine the lowest concentration of target to be detected by our modified PCR:CRISPR-Cas assay, a purified amplification product of 564 bp was titrated. This 564-bp PCR product was obtained by replacing the forward primer (5'-GGCACCAGATTCAACTTTCAAG-3'), and contains the target region for the crRNA and the primers sets to amplify the 282-bp $bla_{OXA-1}$ target sequence. The template titration curve was set in moles. A 12-point target curve was prepared by serial dilution at a factor of 1:5 from $2.0 \times 10^3$ to $4.10 \times 10^{-5}$ attomoles per reaction. The amplification reactions were prepared as described in the previous sections, in a final volume of 20 µL. Amplification was performed using the DreamTaq PCR Master Mix 2X Green under the following amplification conditions, 95°C for 3 min, followed by 30 cycles of 95°C for 1 s, 60°C for 1 s with a TRR of 2.2°C/s, followed by 72°C for 5 min. The Limit of Detection (LoD) was calculated following the formula $LoD = LoB + 2 \times (SD\_low\ concentration\ sample)$ [36], where SD represents the standard deviation of the lowest target concentration that differs from the non-template control reaction.

Meanwhile, amplification using the enzyme produced in the laboratory was performed under the following amplification conditions, 95°C for 3 min, followed by 30 cycles of 95 °C, 60 °C, and 72 °C for 10 s each with a TRR of 2.2°C/s, followed by 72°C for 5 min. The amplification products were visualized by electrophoresis (80 V for 45 min) in agarose gel (1.7%), using a pre-staining method by mixing the tracking buffer with SYBR Gold. In addition, 5 µL of the amplification products were used for fluorescent detection using CRISPR-Cas.

**Spike-in pilot test.** To evaluate the performance of the assay with simulated samples, a spike-in assay was performed on chicken fecal samples, following a modified protocol from [37]. Fecal samples were obtained from a broiler farm for human consumption, located in Lurín, Lima (12°57'13.2"S 76°25'10.7"W). Sample matrix was prepared by mixing 6 g of feces with 60 mL of distilled water. A $bla_{OXA-1}$-positive *E. coli* isolate (SRX147221841 from BioProject PRJNA821865) was grown in LB medium at 37 °C for 18 h at 150 rpm. The absorbance was then measured at 600 nm ($OD_{600}$) and adjusted to a value of 0.05. Eight serial decimal dilutions were then prepared in PBS. From each dilution, 50 µL were spreaded onto LB plates to estimate colony-forming units (CFU/mL). After incubation (37°C for 16 hours), CFU counts were recorded, and dilutions corresponding to 320 and 2400 CFU/ml were selected for a 4-replicate spiking assay. From each selected dilution, 100 µL was transferred to 900 µL of fecal suspension. A volume of 250 µL from each suspension was then used for total DNA extraction, representing to 8 and 60 total CFUs, respectively. These values correspond to an expected bacterial input to the DNA extraction process. Extraction was performed using the ZymoBIOMICS™ DNA Miniprep Kit (Cat. N° D4300, Zymo Research), following the manufacturer's protocol. DNA quantification was performed by absorbance at 260 nm in a Nanodrop One (ThermoFisher, USA). Amplification was performed using the modified PCR with the DreamTaq Green PCR Master Mix (2X), according to the conditions described in the previous sections. Finally, the amplification products were detected using the CRISPR-Cas reaction.

## Results

### Bioinformatic design

Target sequence was analyzed within the $bla_{OXA-1}$ gene family and found to be highly conserved, with only 4.9% single-nucleotide polymorphisms (SNPs) (41/831 nucleotides). Nucleotide variability was mainly concentrated in three regions:

180−200 bp, 290−320 bp, and 560−630 bp (S1 File in S1 Text). The forward primer hybridized within a complete SNP-free region. In contrast, the reverse primer showed a single mismatch in two of three of the $bla_{OXA-47}$ sequences retrieved. Nevertheless, for both forward and reverse primers, annealing of the 3′ bases, essential for the initiation of DNA polymerization, did not show any mismatches (S1 File in S1 Text). Concerning the crRNA, only one mismatch located at the position 4 of the 5′ end was observed in the following variants, $bla_{OXA-47}$, $bla_{OXA-392}$, $bla_{OXA-675}$, and $bla_{OXA-1370}$ (S1 File in S1 Text). As this mismatch is located within what is considered the seed region (i.e., first 5-nt of the crRNA sequence following the PAM site), it could potentially affect the detection of the aforementioned $bla_{OXA}$ variants compared with the other members of the $bla_{OXA-1}$ family. However, this remains to be evaluated, since in this study samples harboring those variants were not assessed.

Specificity of the designed crRNA was analyzed against other $bla_{OXA}$ groups (n = 42). The analysis showed that within the seed sequence, positions +1, +4 and +5 were highly diverse between $bla_{OXA-1}$ family and the other $bla_{OXA}$ groups (Fig 1A). In contrast, positions +2 and +3 were more conserved across all evaluated groups, with cytosine being the most frequent nucleotide (76% and 96%, respectively). Beyond the seed region, the $bla_{OXA-24}$ group showed full-length alignment with the crRNA; however, six mismatches were present along the sequence, resulting in 75% identity. Importantly, a canonical PAM (5′ TTTN 3′) was not observed upstream, and the primers did not align properly for target amplification. Taking together, the findings suggest that the detection of $bla_{OXA-24}$ by our crRNA design is unlikely. Other $bla_{OXA}$ families like $bla_{OXA-10}$, $bla_{OXA-114}$ $bla_{OXA-258}$, and $bla_{OXA-364}$ present a PAM sequence upstream of the aligned region. Nevertheless, multiple mismatches were present, with a mean identity of 55.21%. Among the remaining groups, 39.53% showed partial alignment with the crRNA with identity percentage ranging between 54.17% to 70.83%. Remarkably, none of these alignments present a PAM region upstream of the crRNA binding site. In addition, for all evaluated $bla_{OXA}$ groups, the primers failed to align properly with the respective target sequences. Instead, reverse primer orientation or low coverage annealing

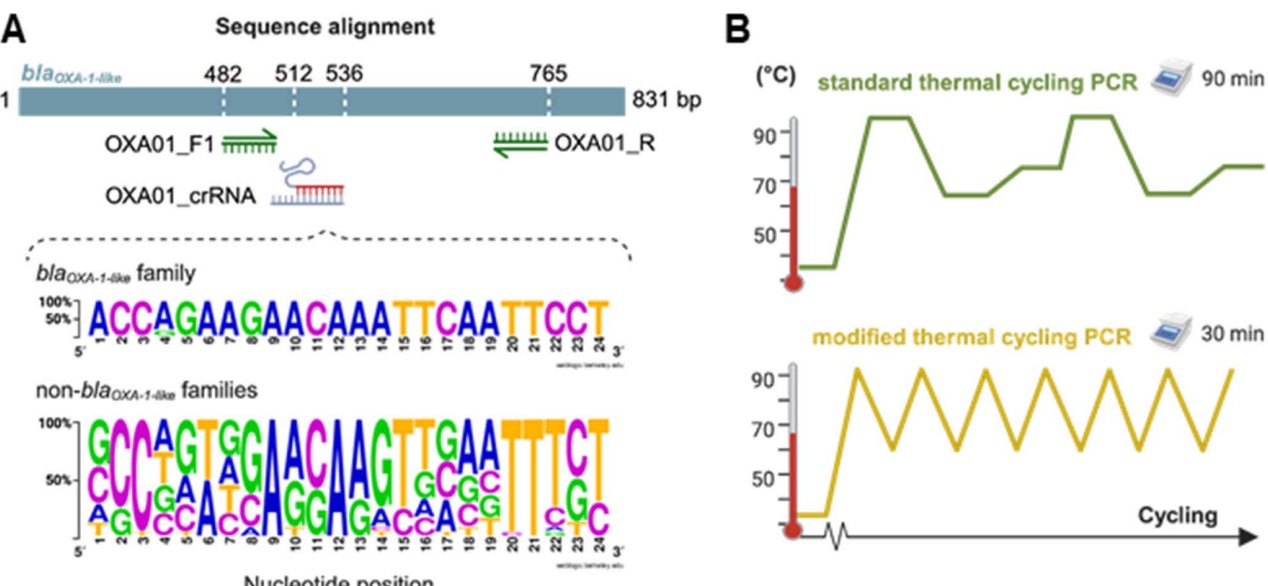

**Fig 1. Study approach highlights. (A)** Schematic representation of primers and crRNA hybridization regions in the $bla_{OXA-1}$ target gene. For the target-specific crRNA sequence, the DNA sequence logo shows the positions and nucleotide variations observed between the $bla_{OXA-1}$-like (n = 46) and other $bla_{OXA}$ families (n = 25). Nucleotide size is related to the frequency (%) in each position (Y-axis). **(B)** Schematic representation of the modified thermal cycling protocol (bottom) compared to conventional PCR (top). Holding times and thermal ramp rate values were modified to avoid the typical 3-step during amplification, while maintaining the denaturation and hybridization temperatures.

was observed, suggesting primer specificity for the bla$_{OXA-1}$-like sequences (S2 File in S1 Text). Finally. Several groups, including bla$_{OXA-21}$, bla$_{OXA-23}$, bla$_{OXA-48}$, bla$_{OXA-50}$, bla$_{OXA-51}$, bla$_{OXA-55}$, bla$_{OXA-58}$, bla$_{OXA-60}$, bla$_{OXA-61}$, bla$_{OXA-62}$, bla$_{OXA-63}$, bla$_{OXA-134}$, bla$_{OXA-143}$, bla$_{OXA-184}$, bla$_{OXA-198}$, bla$_{OXA-213}$, bla$_{OXA-266}$, bla$_{OXA-372}$, bla$_{OXA-548}$, and bla$_{OXA-679}$ did not align with the crRNA (S2 File in S1 Text). In summary, the bioinformatic analysis reveals a high level of specificity of the designed crRNA and primers for the bla$_{OXA-1-like}$ family.

The selected crRNA showed a low ΔG value (−2.99 kcal/mol) for its secondary structure formation, maintaining the scaffold conformation for the Cas interaction. Also, the target-specific region did not show self-complementary, and off-target sites against the human genome.

## Target amplification using a modified thermal cycling protocol

An ideal detection method should achieve a balance between speed and result reliability. Most molecular detection assays depend on target amplification assays. While standard PCR is competitive in terms of sensitivity, it still requires ~90 min. Modifying the thermal cycling can reduce the total amplification time to ~30 min (Fig 1B), by increasing the thermal ramp rate (TRR) and reducing the holding times [38]. The TRR represents the speed at which the thermocycler heats up and cools down between PCR steps, which is inversely proportional to the time.

Considering the technical capabilities of the thermal cycler used (T100; BIO-RAD, USA), ramp rates ranging from 0.6 to 2.2 °C/s were tested. Higher ramp rates resulted in shorter overall reaction times. Notably, a 30-minute PCR using the highest TRR (2.2 °C/s) and holding times of 1 s amplified comparably to the conventional PCR, as positive control (Fig 2A). Based on these results, this value was selected for subsequent assays, as higher values were not evaluated due to minimally expected improvements in time efficiency. To further assess the method's robustness, a less processive, in-house-produced DNA Taq polymerase was evaluated under the same TRR conditions. Given the enzyme's lower pro-cessivity and polymerization rate (nt/s), each PCR step holding-time was extended to 10 s instead of 1 s, based on a prior time optimization curve (S2 Fig in S1 Text). This adjustment increased the total assay time by 14 minutes but still resulted in comparable amplification across all TRR values tested (Fig 2B).

Another variable considered during standardization was the product size as longer products require longer extension (or hold) time. In all assays shown, a 282-bp amplicon size was the final amplification product. Nonetheless, a longer product of 564 bp was also tested. The TRR gradient yielded comparable results using the commercial enzyme (S3 Fig in S1 Text), maintaining the overall turnaround time.

## CRISPR-Cas assay for detection of amplified ARG targets

CRISPR-Cas systems have been used to detect DNA or RNA targets under different types of readouts. Here a fluorescent CRISPR-Cas-based readout was coupled to the modified PCR to detect the bla$_{OXA-1}$ gene in the *E. coli* genome. The selected crRNA in the CRISPR-Cas reaction showed a high target discrimination when genomic DNA was amplified using the modified PCR. Target detection was observed from 10 min onwards, reaching a normalized fluorescent ratio of 10 a.u. at 20 min of assay reading (Fig 3). Direct fluorescent visualization also allowed discrimination between positive and negative reactions (Fig 3A). A cutoff value of 1.92 a.u. at 20 min was estimated for the assay under optimized conditions (S4 Fig in S1 Text). The assay showed full specificity for bla$_{OXA-1}$ sequence, when compared to other beta-lactamases (S5 Fig in S1 Text).

The modified PCR in combination with the fluorescent CRISPR-Cas reaction was able to detect 5.12 x 10$^{-3}$ attomoles of a linear target per reaction, yielding a normalized fluorescence mean (NF$_{ntc}$) value of 2.07, which represents a mini-mal increase over the determined cutoff. By contrast, a 5-fold higher target quantity (2.56 x 10$^{-2}$ attomoles per reaction) produced an NF$_{ntc}$ value of 5.88, corresponding to approximately 2-fold higher signal relative to the cutoff. Accordingly, the LoD was estimated to be 5 x 10$^{-3}$ attomoles, consistent with the lowest target quantity that differs from the blank. For better comparison, this target quantity is equivalent to 3000 gene copies (Fig 3B).

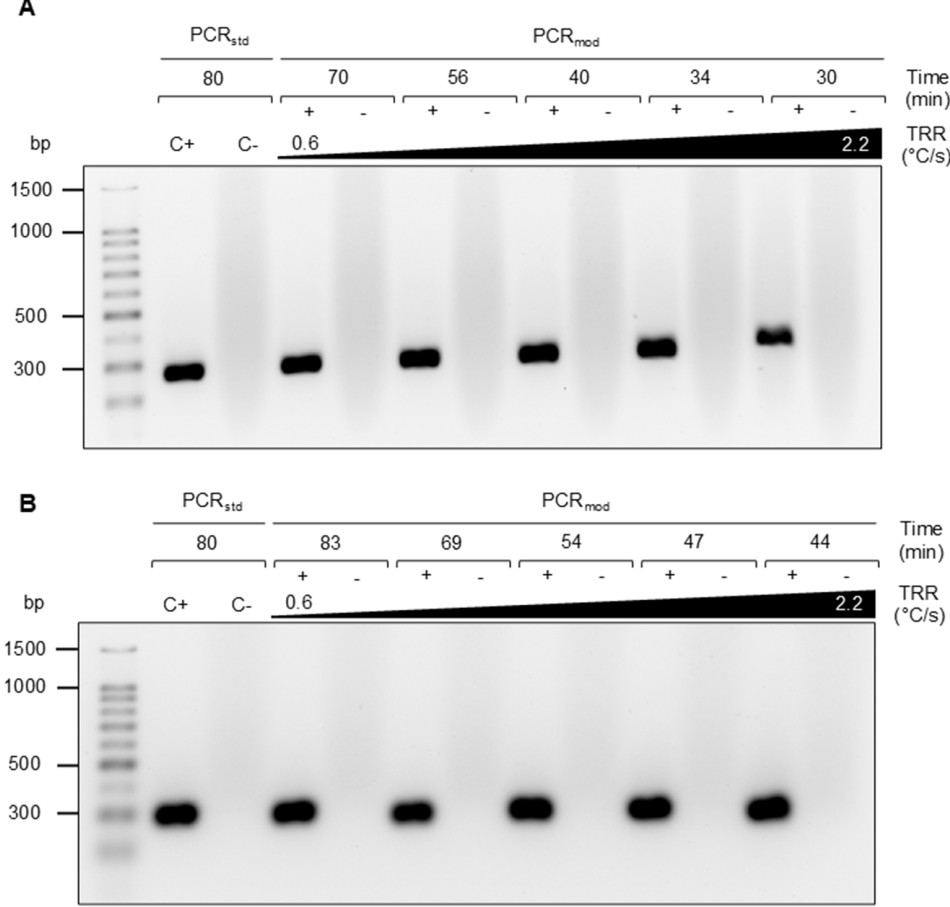

**Fig 2. Amplification products obtained using a PCR with modified thermal cycling (PCR$_{mod}$).** Amplified 282-bp sequences were visualized using an agarose gel (1.7%) alongside a 100-bp ladder. Standard PCR (PCR$_{std}$) was used as a positive amplification reaction (C+). **(A)** DNA amplification using a commercial DNA Taq polymerase using different TRR values: 0.6, 0.8, 1.2, 1.6, and 2.2 °C/s. After each ramp 1 s was used as hold or extension time. Total PCR time is indicated for each TRR. Each condition was tested also in the absence of the template DNA (-). **(B)** As A using a locally produced DNA Taq polymerase and 10 s for hold or extension time.

To further evaluate the assay performance, a locally produced Taq DNA polymerase was tested with the modified amplification cycling. Compared to the modified PCR amplification with the commercial polymerase, the in-house enzyme detects 600-fold lower target concentration. However, when amplification protocols were compared using only the in-house enzyme, minimal differences were observed (Fig 3C, mid and right lane). The lowest target quantity detected with the locally produced enzyme was 3.2 attomoles per reaction, regardless of the amplification protocol. However, a 2-fold higher signal was observed for the standard PCR amplification (Fig 3C).

Finally, bacterial cell detection was evaluated by spiking poultry feces samples. Using a commercial DNA Taq polymerase, the optimized assay enables detecting 60 CFU, yielding a mean NF$_{ntc}$ value of 10.85. In contrast, a 7-fold lower CFU produced only a marginal increase over the negative control, reaching a mean NF$_{ntc}$ value of 2.23. Taken together, our results demonstrate that the described assay is competitive in terms of analytical sensitivity for detecting biological markers such as antimicrobial resistance genes even in simulated samples.

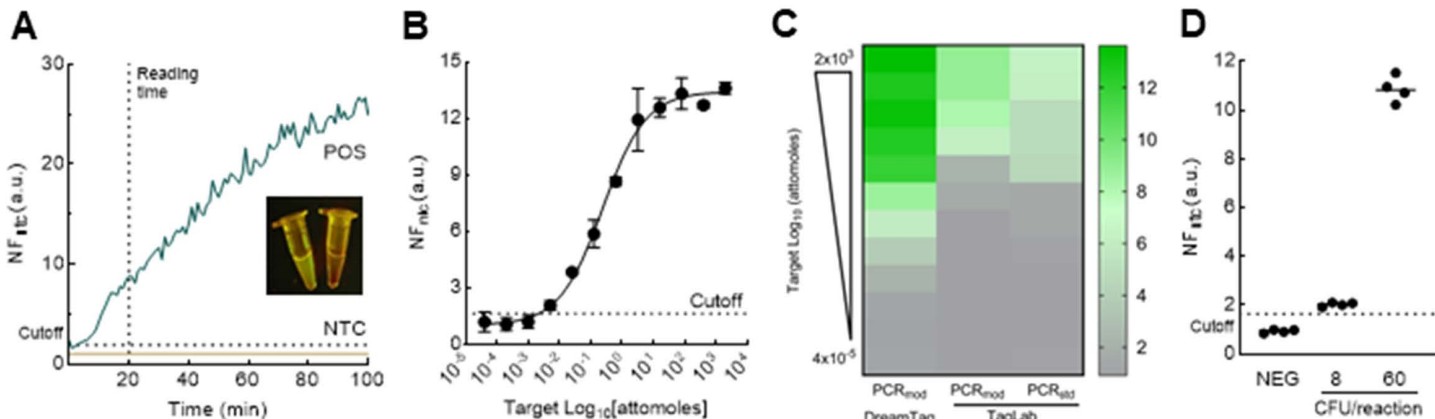

**Fig 3. Assay optimization and performance evaluation. (A)** Discrimination assay ability. Normalized fluorescent ratio values ($NF_{ntc}$) are plotted over time for a positive sample (POS) and a non-template control (NTC). The inset image shows the fluorescence of the POS and NTC reactions as observed with a blue-light transilluminator. **(B)** Analytical sensitivity of the optimized assay. The assay response to decreasing target quantity is plotted as $NF_{ntc}$ values over target quantity expressed in logarithmic scale of total moles of a purified 564-bp amplicon. Error bars indicate the SEM value from the mean from duplicates. **(C)** Thermal cycling performance comparison. The heatmap shows the target detection performance between reactions using a commercial DNA polymerase (DreamTaq) in the modified PCR protocol ($PCR_{mod}$) (left lane), a locally produced polymerase (TaqLab – mid and right late) in the modified PCR protocol (midlane), and in a standard PCR protocol ($PCR_{std}$) (right lane). Color intensity corresponds to the $NF_{ntc}$ values at each DNA template concentration that was evaluated (N = 36). **(D)** Spiking assay using a $bla_{OXA-1}$-positive *E. coli* in a poultry feces sample. The assay was evaluated against a non-spiked control (NEG) and samples spiked with 8 or 60 CFU per reaction (n = 4). The horizontal line in each group represents the mean $NF_{ntc}$ value. For all panels, the dotted line in the Y-axis represents the $NF_{ntc}$ cutoff to discriminate positive from negative samples at 20 min. The dotted line in the X-axis represents the selected reading time for the CRISPR-Cas assay.

## Discussion

Extended- and narrow-spectrum β-lactamases remain a major concern in clinical microbiology due to their prevalence, diversity, and global distribution [2,39]. Among these, the $bla_{OXA-1}$-like gene has been reported in multiple bacterial families, including *Enterobacteriaceae*, *Moraxellaceae*, and *Pseudomonadaceae* [40], which include most members of the ESKAPE group. Recent studies highlight high frequencies exceeding 50% in clinical *Klebsiella pneumonia* isolates [4], and its identification as the most abundant plasmid-borne blaOXA gene (57.93%) among *E.coli* isolates [5] regardless of geographical region. The $bla_{OXA-1}$ gene is frequently associated with class 1 integrons and often co-occurs with other resistance determinants, such as aminoglycoside-resistance genes like aac(6')-lb-cr, other carbapenemases, and even colistin-associated resistance genes [5]. These findings highlight the clinical and epidemiological significance of the $bla_{OXA}$ gene families.

In this study, we developed and evaluated a rapid molecular assay for $bla_{OXA-1}$-like gene detection, combining a faster PCR protocol with CRISPR-Cas fluorescence readout. With this approach, amplification time was reduced to 30 min, representing a 62.5% decrease compared to conventional PCR. Thus, the complete workflow, from amplification to CRISPR-Cas detection, is completed in 50 minutes. With a commercial DNA Taq polymerase, the assay achieved a detection limit of 5 x 10$^{-3}$ attomoles of pure DNA. When tested with simulated samples, it reliably detected ≥60 CFU per reaction, with a marginal signal at 8 CFU per reaction, equivalent to two-fold above negative controls. It is important to emphasize the value of evaluating assay sensitivity using metrics that reflect real-world conditions, such as CFU numbers. Since ARGs are primarily disseminated through bacterial hosts in clinical and environmental settings, biological titters provide a more relevant assessment of detection performance than purified DNA concentrations alone.

CRISPR-Cas-based assays are characterized by a two-level specificity, one determined by primer annealing, and the other by the crRNA binding. Although assay specificity was not evaluated *in vitro*, our *in silico* analysis indicated a high expected specificity. LbCas12a can tolerate mismatches depending on their distance from the PAM. The first five

nucleotides adjacent to the PAM constitute the seed sequence, which is essential for Cas12a activity [41]. Mismatches close to the PAM have stronger effects; for example, a mismatch at position +1 abolishes target recognition, whereas a mutation at position +5 partially reduces template recognition [42]. After position +5, nucleotide variation is generally tolerated; however, at least 17 out of 22 nucleotides (77%) must match for Cas12a-mediated cleavage to occur. $bla_{OXA}$ groups other than group 1 did not exhibit such level of complementarity with either the designed crRNA or the primers, thus it is expected that our system should not detect them.

Our results compare favorably with other CRISPR-Cas-based β-lactamase detection assays. Recently, a PCR–Cas12a system for $bla_{OXA-1}$ was reported, with a 70-minute turnaround (65 minutes amplification followed by 5 minutes detection) and a limit of 1.25 gene copies [40]. However, their assay differs from ours in several reaction parameters, including temperature (25 °C vs. 37 °C), reaction buffer (NEB r2.1 vs. Holmes), and fluorescent probe design (TTATT vs. 8C or hairpin ssDNA), exemplifying the diversity of published assay protocols. Other studies have targeted $bla_{GES}$, $bla_{NDM-1}$, $bla_{OXA-23}$, $bla_{OXA-41}$, and $bla_{OXA-48}$ using isothermal amplification (RPA, LAMP) with LbCas12a or LwaCas13a, reporting turnaround times of 60–120 minutes and diverse limits of detection such as $1.3 \times 10^{-6}$ ng/µL [43], 2.7 CFU/ml [44], and species-dependence range such as $10^3$–$10^7$ CFU/mL [45]. Such methodological variability complicates direct performance comparisons, yet our assay is competitive in terms of speed, sensitivity, and resource requirements.

From an implementation perspective, conventional PCR remains widely used albeit requiring a thermocycler and longer run times. Isothermal methods such as LAMP and RPA offer faster amplification without thermal cycling but can suffer from nonspecific amplification [46], limited reagent availability [47], and insufficient validation in clinical matrices [48]. Unlike current molecular methods for detecting $bla_{OXA}$ genes, such as PCR, qPCR or NGS, which are time-consuming or require specialized equipment, our CRISPR-Cas-based assay provides a rapid fluorescent readout compatible with low-cost, portable tools such as blue-light transilluminator or 3D-printed fluorometers, bypassing the necessity of expensive and additional instrumentation such sequencers, real-time thermocyclers, or electrophoresis chambers, enabling future democratization of CRISPR-Cas-based molecular detection assays.

Although the principal limitation of our study is the absence of real sample validation, our approach still offers a balanced alternative, maintaining high analytical sensitivity while reducing turnaround time and relying on equipment common to most basic molecular laboratories. Our system could find a compelling advantage against traditional molecular detection platforms if applied for environmental monitoring of ARG due to its velocity, low cost, and compatibility with minimal equipment.

When using an in-house-produced DNA Taq polymerase, amplification time increased by 15 min, due to the need for ≥ 10-s holds at each temperature step during amplification. This may reflect suboptimal buffer composition or reagent concentrations. Sensitivity was also reduced, with a detection limit of 3.2 attomoles—over 600-fold less sensitive than the commercial enzyme. However, when comparing both PCR amplification protocols (e.g., standard *vs* modified thermal cycling) with the in-house enzyme, the detection limit differed by only five-fold, consistent with Chen's findings that amplification strategy may have less impact than enzyme quality [38]. While further optimization is needed, the use of locally produced enzymes remains a strategic option for mitigating supply chain disruptions, as seen during the COVID-19 pandemic [49].

In conclusion, the proposed assay expands the toolkit for ARG gene detection, particularly in low-resource settings where basic molecular equipment is available. Its combination of speed, sensitivity, and adaptability makes it a promising candidate for further validation.

## Conclusion

We proposed a proof-of-concept assay that combines modified thermal cycling–PCR with fluorescent CRISPR-Cas detection. This approach significantly reduces the total turnaround time for detecting antimicrobial resistance genes using basic molecular biology equipment. The assay, when using commercially available enzymes, retains competitive analytical sensitivity. However, further validation is required to confirm its diagnostic value.

## Supporting information

**S1 File. Multiple sequence alignment of the DNA sequences retrieved from the CARD and NDARO databases related to the blaOXA-1 gene.**
(ALN)

**S2 File. Multiple sequence alignments of the DNA sequences retrieved from the CARD and NDARO databases related to the non-blaOXA-1 groups.**
(ZIP)

**S1_raw_images. Original uncropped and unadjusted images of agarose gels.**
(PDF)

**S1 Supporting Information. Supplementary figures.**
(DOCX)

## Acknowledgments

We are very thankful to Dr. Maya Nadimpalli for providing access to the *E. coli* isolate sample used in this study, and Dr. Pohl Milon for his valuable insights for the manuscription revision.

## Author contributions

**Conceptualization:** Maryhory Vargas-Reyes, Roberto Alcántara.

**Formal analysis:** Ana Quiroz-Huanca, Maryhory Vargas-Reyes, Kiara Flores-Jimenez.

**Funding acquisition:** Roberto Alcántara.

**Investigation:** Ana Quiroz-Huanca, Juan Diego López, Kiara Flores-Jimenez, Sofia Saldarriaga-Morán, Karla Cifuentes.

**Methodology:** Maryhory Vargas-Reyes, Roberto Alcántara.

**Supervision:** Roberto Alcántara.

**Visualization:** Roberto Alcántara.

**Writing – original draft:** Ana Quiroz-Huanca, Maryhory Vargas-Reyes, Juan Diego López, Sofia Saldarriaga-Morán, Karla Cifuentes, Roberto Alcántara.

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
