## [Decision Letter · Decision Letter 0]

3 Feb 2026

PONE-D-25-60550Thermal optimized PCR coupled to CRISPR-Cas12a for rapid detection of blaOXA-1 resistance genePLOS One

Dear Dr. Alcántara,

Thank you for submitting your manuscript to PLOS ONE. After careful consideration, we feel that it has merit but does not fully meet PLOS ONE’s publication criteria as it currently stands. Therefore, we invite you to submit a revised version of the manuscript that addresses the points raised during the review process.

 The reviewer has pointed out a number of areas in which revision is needed. Please addess all reviewer comments point by point.==============================

We look forward to receiving your revised manuscript.

Kind regards,

Iddya Karunasagar

Academic Editor

PLOS One

Journal Requirements:

“This research was supported by a research grant from the Universidad Peruana de Ciencias Aplicadas with number C-004-2021-1 awarded to Roberto Alcántara.”

“This research was supported by a research grant from the Universidad Peruana de Ciencias Aplicadas with number C-004-2021-1 awarded to Roberto Alcántara.”

5. Please be informed that funding information should not appear in the Acknowledgments section or other areas of your manuscript. We will only publish funding information present in the Funding Statement section of the online submission form. Please remove any funding-related text from the manuscript.

Additional Editor Comments (if provided):

Please see reviewer comments and revise the manuscript accordingly.

Reviewers' comments:

Reviewer's Responses to Questions

**Comments to the Author**

1. Is the manuscript technically sound, and do the data support the conclusions?

Reviewer #1: Yes

Reviewer #2: Yes

2. Has the statistical analysis been performed appropriately and rigorously? 

Reviewer #1: I Don't Know

Reviewer #2: Yes

3. Have the authors made all data underlying the findings in their manuscript fully available?

Reviewer #1: Yes

Reviewer #2: Yes

4. Is the manuscript presented in an intelligible fashion and written in standard English?

Reviewer #1: Yes

Reviewer #2: Yes

5. Review Comments to the Author

Reviewer #1: Thermal optimized PCR coupled to CRISPR-1 Cas12a for rapid detection of blaOXA-1 resistance gene

This study developed a CRISPR-Cas12a-based fluorescence method for the detection of blaOXA-1 gene, with a sensitivity of 8 CFU. Primers designed from consensus sequences of blaOXA-1 were employed in conventional PCR for initial optimization of conditions, followed by CRISPR-Cas detection.

Major comments

The study does not convincingly prove the specificity of the assay. The authors should have tested their primers with diverse blaOXA-1-harbouring isolates, both E. coli and other Enterobacterales, and other ESBL- and MBL-encoding genes to prove the specificity of the method developed.

What is the specific advantage of this method, when PCR methods for blaOXA variants are already available?

Further, the authors need to clearly define the goal (s) and the application of their work in the abstract and in the introduction section. It is only evident after section 2.4.3. (Spike-in pilot test) that the method was indeed designed to detect the target organisms (E. coli carrying blaOXA-1 gene) in chicken feces.

In L217-223: It is stated here that the mismatches in primer binding sites might potentially affect detection of the variants of blaOXA gene. This being the case, is this method specific to only blaOXA-1 gene? Does this not limit the application of this method, since blaOXA genes are very diverse?

L165: Here, amplification product of 564 bp has been mentioned. What does this product refer to?

5 x 10-3 attomoles of pure DNA: Please compare and discuss the sensitivity with pure DNA and the spiked bacterial cells (8 CFU/reaction). Alternatively, this may be represented as gene copies.

Minor comments

Please label A & B in Figure 2

Figure 3. Please move (A) to the next line

Reviewer #2: The study is simple yet performed with relevant outcomes. However, authors can address the following comments

1. What is the inclusion/exclusion criteria used while selecting 46 blaOXA-1 gene sequences?

2. Authors need to specify the version of tools used and parameter settings wherever applicable throughout the methodology

3. Times reported for different TRR with lab-made Taq (e.g., 0.6 °C/s = 83 minutes) differ from those with DreamTaq. Can authors clarify why total run times differ, given similar cycle numbers and temperatures?

4. Can authors provide alternative gel image for Figure 2A since the bands appear at different size range (could be due to imaging error)

5. Two different plate readers were used. Were the instruments cross-calibrated to ensure comparable fluorescence readings?

6. How was 10 blaOXA-1 negative E. coli isolates confirmed? Whether by PCR or WGS?

7. Authors used a 564 bp purified amplification product as target for Target concentration curve. Why is this length different from the 282 bp region described earlier?

8. For amplification with lab-made enzyme cycle time was 10 s per step, while 1 s protocol was followed for DreamTaq. Can authors justify the difference?

9. During spike test, OD600 was adjusted to 0.05 before serial dilutions. Was CFU conversion calibrated for this isolate or assumed from literature?

10. “From each selected dilution, 100 μL were transferred to 900 μL of the fecal suspension, then, 250 μL of each suspension was used for total DNA extraction, corresponding to an estimated 8 and 60 total CFUs, respectively.” Was extraction efficiency (recovery of DNA per CFU) evaluated or assumed?

11. “Extraction was performed using the ZymoBIOMICS™ DNA Miniprep Kit (Cat. N° D4300, Zymo Research), following the manufacturer's protocol.” Could authors compare kit-based extraction to simpler or cheaper methods that may be relevant for field deployment?

12. For spiked samples, authors used the modified PCR with DreamTaq and subsequent CRISPR detection. Did authors also test lab-produced Taq in the fecal matrix, and if not, why?

6. PLOS authors have the option to publish the peer review history of their article (what does this mean?). If published, this will include your full peer review and any attached files.

Reviewer #1: No

Reviewer #2: No

---

## [Author Response · Author response to Decision Letter 1]

3 Mar 2026

Thank you for your comments and suggestions. We have carefully considered each requirement and implemented the changes that enhance the quality of the manuscript.

---

## [Decision Letter · Decision Letter 1]

8 Apr 2026

PONE-D-25-60550R1Thermal optimized PCR coupled to CRISPR-Cas12a for rapid detection of blaOXA-1 resistance genePLOS One

Dear Dr. Alcántara,

Thank you for submitting your manuscript to PLOS ONE. After careful consideration, we feel that it has merit but does not fully meet PLOS ONE’s publication criteria as it currently stands. Therefore, we invite you to submit a revised version of the manuscript that addresses the points raised during the review process.

Some minor changes in language improvement, formattig needed

We look forward to receiving your revised manuscript.

Kind regards,

Iddya Karunasagar

Academic Editor

PLOS One

Journal Requirements:

Additional Editor Comments :

Please see some comments from reviewer regarding improvements needed.

Reviewers' comments:

Reviewer's Responses to Questions

**Comments to the Author**

1. If the authors have adequately addressed your comments raised in a previous round of review and you feel that this manuscript is now acceptable for publication, you may indicate that here to bypass the “Comments to the Author” section, enter your conflict of interest statement in the “Confidential to Editor” section, and submit your "Accept" recommendation.

Reviewer #1: All comments have been addressed

Reviewer #2: All comments have been addressed

2. Is the manuscript technically sound, and do the data support the conclusions?

Reviewer #1: Yes

Reviewer #2: Yes

3. Has the statistical analysis been performed appropriately and rigorously? 

Reviewer #1: Yes

Reviewer #2: Yes

4. Have the authors made all data underlying the findings in their manuscript fully available?

Reviewer #1: Yes

Reviewer #2: Yes

5. Is the manuscript presented in an intelligible fashion and written in standard English?

Reviewer #1: Yes

Reviewer #2: Yes

6. Review Comments to the Author

Reviewer #1: The authors have responded to the queries raised in the previous review. Yet, the application of blaOXA-1 group -specific detection method has limited applicability in view of the fact that this does not cover the most common blaOXA gene(s) either in E. coli or other Gram-negative pathogens such as Acinetobacter baumannii.

The application of an in-house polymerase is something unique in this study, although it is not clear how the results obtained using an in-house polymerase can be reproduced in other laboratories beyond the geographic region where this study was performed. Further, the authors have stated in the manuscript that this polymerase had a lower efficiency (processivity) compared to commercial polymerase (s). In my opinion, the section describing in-house polymerase may be removed, as the data/results are relevant only to the laboratory employing it.

A few suggestions are below for authors’ consideration

75: Please correct as ..”needing long amplification times.

L96-97: Please rephrase this sentence.

Please write blaOXA correctly throughout the manuscript. OXA should be subscript of bla and in non-italics.

L133: The dNTP concentration of 0.5 nM seems to be too low. Please check. Similarly, the Taq concentration is generally stated in units, but you have indicated in ng/µL.

L163-164: Please write bla genes in correct format.

L208: Please provide the strain number/designation of blaOXA-1 isolate you used in this study

L208: Replace “cultivated” with “grown”

L211: Seeded? It appears that you performed a spread plate. However, spread plate method employs 100 microliters, not 50.

L214-216; Please rephrase the sentence.

L446-451: Agreed that qPCR and NGS are expensive. In what way your method is less expensive or technically less demanding than conventional PCR?. This method also requires PCR amplification, followed by fluorescence detection.

Reviewer #2: (No Response)

7. PLOS authors have the option to publish the peer review history of their article (what does this mean?). If published, this will include your full peer review and any attached files.

Reviewer #1: No

Reviewer #2: No

---

## [Author Response · Author response to Decision Letter 2]

28 Apr 2026

Minor changes have been resolved as requested.

Regarding to the Reviewer #1 comments:

"The authors have responded to the queries raised in the previous review. Yet, the application of blaOXA-1 group -specific detection method has limited applicability in view of the fact that this does not cover the most common blaOXA gene(s) either in E. coli or other Gram-negative pathogens such as Acinetobacter baumannii."

Thank you for this perspective. We agree that blaOXA genes are remarkably diverse and that different variants are more prevalent depending of the host species, geographical area and clinical/environmental context. However, we maintain that blaOXA-1 represents a priority target for the following reasons:

• High prevalence in other clinically relevant Gram-negative: Recent literature confirms high frequencies exceeding 50% in clinical Klebsiella pneumoniae isolates (Meng et al., 2023), and identifying blaOXA-1-carrying plasmids as the most abundant (57.93%) in E. coli isolates (Wu et al., 2024). In both studies, independently of the geographic area.

• Co-presence with other ARG markers: blaOXA-1 is frequently associated with class 1 integrons and co-occurs with other relevant resistance markers such as blaXTX-M-15 and aac(6’)-Ib-cr.

• Presence in environmental and food samples: Some studies have reported high prevalence of blaOXA-1 in soil and manure samples, 45.16% (Laconi et al., 2021), and in E.coli isolated from poultry meat, 40.54% (Ali 2025).

Finally, we acknowledge that this assay does not cover all relevant and high frequent OXA-carbapanemases. However, as a proof-of-concept, we intended to develop a rapid, user-friendly and affordable tool for monitoring this emerging resistance gene.

These statements and references have been added to the manuscript.

"The application of an in-house polymerase is something unique in this study, although it is not clear how the results obtained using an in-house polymerase can be reproduced in other laboratories beyond the geographic region where this study was performed. Further, the authors have stated in the manuscript that this polymerase had a lower efficiency (processivity) compared to commercial polymerase (s). In my opinion, the section describing in-house polymerase may be removed, as the data/results are relevant only to the laboratory employing it."

Thank you for your perspective. We maintain that including the local-produced DNA polymerase results is important for contributing to the democratization of molecular diagnostics. By providing full standardized protocols (Reference 30) and evidence of their implementation, we demonstrate that molecular detection can be achieved based on local efforts. Although the Taq DNA polymerase showed a lower performance compared to a commercial alternative, its inclusion serves as a proof-of-concept for local enzyme production. We stated that further optimization is needed and does not discard the utility of local-produced enzymes for implementing molecular detection assays, even in emergency contexts such as future pandemics as we stated in lines 468-470.

---

## [Decision Letter · Decision Letter 2]

3 May 2026

Thermal optimized PCR coupled to CRISPR-Cas12a for rapid detection of blaOXA-1 resistance gene

PONE-D-25-60550R2

Dear Dr. Alcántara,

We’re pleased to inform you that your manuscript has been judged scientifically suitable for publication and will be formally accepted for publication once it meets all outstanding technical requirements.

Kind regards,

Iddya Karunasagar

Academic Editor

PLOS One

Additional Editor Comments (optional):

All reviewer comments have been addressed.

Reviewers' comments:

Reviewer's Responses to Questions

**Comments to the Author**

1. If the authors have adequately addressed your comments raised in a previous round of review and you feel that this manuscript is now acceptable for publication, you may indicate that here to bypass the “Comments to the Author” section, enter your conflict of interest statement in the “Confidential to Editor” section, and submit your "Accept" recommendation.

Reviewer #1: All comments have been addressed

2. Is the manuscript technically sound, and do the data support the conclusions?

Reviewer #1: Yes

3. Has the statistical analysis been performed appropriately and rigorously? 

Reviewer #1: Yes

4. Have the authors made all data underlying the findings in their manuscript fully available?

Reviewer #1: Yes

5. Is the manuscript presented in an intelligible fashion and written in standard English?

Reviewer #1: Yes

6. Review Comments to the Author

Reviewer #1: The authors have satisfactorily addressed all the queries raised in the previous review.

L263-265 & ub Figure legends: Italicize "bla".

Fig 2B legend: "Same as A, using...

Fig. 3B: the x axis lebel say "target log1o attomoles, but the numbers are represented to the power of 10. Please check if this is correct.

7. PLOS authors have the option to publish the peer review history of their article (what does this mean?). If published, this will include your full peer review and any attached files.

Reviewer #1: No

---

## [Editor Report · Acceptance letter]

PONE-D-25-60550R2

PLOS One

Dear Dr. Alcántara,

I'm pleased to inform you that your manuscript has been deemed suitable for publication in PLOS One. Congratulations! Your manuscript is now being handed over to our production team.

Kind regards,

on behalf of

Dr. Iddya Karunasagar

Academic Editor

PLOS One